# RSG-YOLOV8: Detection of rice seed germination rate based on enhanced YOLOv8 and multi-scale attention feature fusion

**Huikang Li**[1], **Longbao Liu**[1], **Qi Li**[1], **Juan Liao**[1], **Lu Liu**[1], **Yujun Zhang**[2], **Qixing Tang**[1]*, **Yuan Rao**[1], **Yanwei Gao**[1]

**1** College of Engineering, Anhui Agricultural University, Hefei, China, **2** Key Laboratory of Environmental Optics and Technology, Chinese Academy of Sciences, Anhui Institute of Optical Precision Machinery, Hefei, Anhui, China

* qxtang@ahau.edu.cn

**Data Availability Statement:** The underlying data are held at Figshare. DOI: 10.6084/m9.figshare. 27288549.

## Abstract

The lack of obvious difference between germinated seeds and non-germinated seeds will cause the low accuracy of detecting rice seed germination rate, remains a challenging issue in the field. In view of this, a new model named Rice Seed Germination-YOLOV8 (RSG-YOLOV8) is proposed in this paper. This model initially incorporates CSPDenseNet to streamline computational processes while preserving accuracy. Furthermore, the BRA, a dynamic and sparse attention mechanism is integrated to highlight critical features while minimizing redundancy. The third advancement is the employment of a structured feature fusion network, based on GFPN, aiming to reconfigure the original Neck component of YOLOv8, thus enabling efficient feature fusion across varying levels. An additional detection head is introduced, improving detection performance through the integration of variable anchor box scales and the optimization of regression losses. This paper also explores the influence of various attention mechanisms, feature fusion techniques, and detection head architectures on the precision of rice seed germination rate detection. Experimental results indicate that RSG-YOLOV8 achieves a $mAP_{50}$ of 0.981, marking a 4% enhancement over the $mAP_{50}$ of YOLOv8 and setting a new benchmark on the RiceSeedGermination dataset for the detection of rice seed germination rate.

## Introduction

Rice is a cornerstone of global food security, acting as a principal staple crop worldwide. The germination rate of rice seeds is a critical indicator for evaluating rice yield potential and is a key metric in seed quality assessment [1]. The diminutive nature and clustered configuration of rice seeds, however, complicate accurate assessment, often leading to decreased precision. Thus, the accurate detection of germinated seeds is paramount in precisely estimating rice yields.

In recent advancements, the confluence of computer hardware improvements and the evolution of computer vision and image processing techniques has facilitated significant progress

**Funding:** This work is supported by Project of National Natural Science Foundation of China (52105539), Anhui Natural Science Foundation (2108085QD179), National Engineering Technology Research Center (2005DP173065-2022-01), Key Laboratory of Agricultural Sensors, Ministry of Agriculture and Rural Affairs, Anhui Provincial Key Laboratory of Smart Agricultural Technology and Equipment, Anhui Agricultural University (KLAS2022KF011). The funders had no role in study design, data collection and analysis, decision to publish, or preparation of the manuscript.

**Competing interests:** The authors have declared that no competing interests exist.

in the assessment of seed germination rate [2, 3]. Zhang et al. unveiled a groundbreaking germination grain counting algorithm based on the premise that the intersection length between the embryo and the grain is shorter than the embryo's circumference. They improved upon the coarse segmentation results from the k-means clustering algorithm through the application of a one-dimensional Gaussian filter and a fifth-degree polynomial for refinement [4]. Tan et al. introduced an algorithm adept at identifying and counting conjoined rice grains. By employing wavelet transform and Gaussian filtering, they enhanced the contrast of grayscale images and reduced noise. Furthermore, they addressed and unified over-segmented regions by deploying an advanced corner detection algorithm, verifying the alignment of segmentation line endpoints with corner points [5].

Moreover, Zhao et al. introduced a sophisticated approach that combines image segmentation, a Transformer encoder, a dedicated small target detection layer, and control distance intersection-over-union (CDIoU) loss to significantly enhance detection accuracy. Their convolutional neural network (CNN) is adept at identifying the germination status of rice seeds and autonomously quantifying the total number of germinated seeds [6]. Predominantly, these studies underscore the importance of exploiting differences in color or length between seeds and embryos to determine germination status. Nonetheless, the similar texture characteristics and size disparities of germinated seeds and non-germinated seeds, can obscure these differences, leading to potential misclassification and reduced accuracy in detection [4].

A comprehensive analysis of the existing literature reveals that although significant progress has been made in the field of target detection, there are still some research gaps. Firstly, the effectiveness of existing models in dealing with multi-scale feature fusion is still limited, especially in the task of target detection in complex contexts. In addition, although multiple attention mechanisms have been proposed, their performance differences in different application scenarios have not been fully explored.

To address the challenges of high accuracy in rice seed germination testing and to fill these gaps, the RSG-YOLOV8, a cutting-edge model is designed to accurately quantify rice seed germination rate. The principal contributions of this research are delineated as follows:

1. Cross Stage Partial DenseNet (CSPDenseNet): The CSPDenseNet is proposed in this work, aimed at augmenting gradient flow and reducing computational demand. Each stage of CSPDenseNet is composed of a partial dense block coupled with a partial transition layer. Unlike the conventional DenseNet, where the base layer's channel count substantially surpasses the growth rate, the CSPDenseNet utilizes a partial dense block. In this configuration, only half of the initial channels contribute to the dense layer operations, addressing nearly half of the computational bottleneck efficiently [7].

2. Bi-level Routing Attention (BRA): The BRA algorithm is introduced to highlight salient features while minimizing redundancy through a dynamic and sparse attention mechanism. By eliminating non-essential key-value pairs at the coarse region level, the BRA ensures the retention of only crucial routing areas, thereby optimizing feature selection [8].

3. Generalized Feature Pyramid Network (GFPN): The GFPN is incorporated in the work, an architecture devised to re-engineer the original 'Neck' component of YOLOv8, facilitating efficacious feature integration across different scales. The GFPN employs dense connections and a Queen-Fusion structure to produce highly integrated features. Furthermore, the Concat operation is utilized over summation for feature fusion, significantly mitigating the risk of information loss [9].

4. Added Detection Head: Addressing the limitations inherent in the original YOLOv8 detection heads for the context of rice seed detection, an innovative detection head is introduced

in this work. This novel component leverages shallow information extracted from the initial C2f (shortcut) module of the input image and integrates it with a supplementary feature fusion network. The introduction of specific anchor box scales and the optimization of regression losses within this new detection head markedly improve detection accuracy. Consequently, this advancement significantly boosts the model's performance, catering to the nuanced demands of rice seed detection with increased precision and effectiveness.

# 1. Rice seed germination rate detection model

**1.1 YOLOv8 model.** This research extends the YOLOv8 framework as delineated in Fig 1, incorporating modifications designed to enhance rice seed detection accuracy. The YOLOv8 architecture is structured around three core components: the Backbone, Neck, and Head. The Backbone consists of Convolution (Conv), C2f (shortcut), and Spatial Pyramid Pooling-Fast (SPPF) modules. The Conv module executes convolution operations on the input images, assisting the C2f module in efficient feature extraction, whereas the SPPF module is instrumental in generating outputs of adaptive sizes [10].

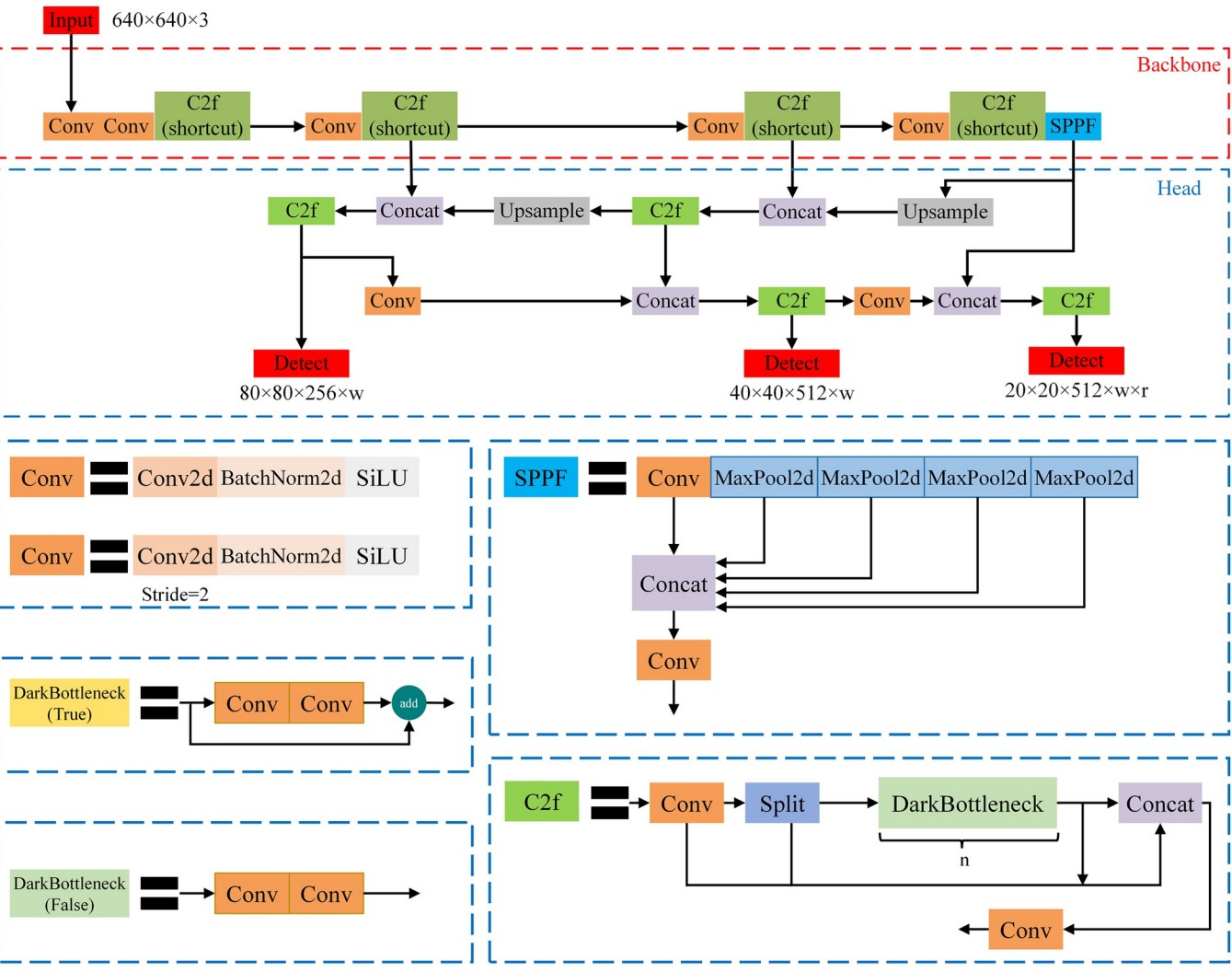

**Fig 1. Overview of YOLOv8 model.**

The Neck, employing the combined architectures of the Feature Pyramid Network (FPN) and Path Aggregation Network (PANet), adeptly extracts and amalgamates features across multiple scales. This integration significantly enhances the model's detection performance and robustness [11]. The Head component, with its decoupled structure, segregates classification and regression tasks into distinct branches, thereby reducing task conflict and fostering improved accuracy.

Notwithstanding its merits, YOLOv8 encounters several challenges. Predominantly, its architecture is characterized by an extensive reliance on convolutional and C2f blocks, escalating computational complexity and the total number of parameters. This complexity, coupled with a constrained detection head capacity, does not fully accommodate the specificities of rice seed detection, particularly in identifying objects beyond the model's original scaling capabilities. Furthermore, there is room for optimization within the Backbone's feature extraction and the Neck's feature fusion processes [12].

In response to these challenges, the present paper proposes a set of modifications to the YOLOv8 model. These enhancements are aimed at optimizing the model's structural efficiency and its effectiveness in the specific application of rice seed detection [13–16].

**1.2 RSG-YOLOV8.** Building upon the advancements of the YOLOv8 model, this paper introduces the RSG-YOLOV8 model, depicted in Fig 2. The proposed model incorporates several key modifications aimed at enhancing both efficiency and accuracy. The integration of CSPDenseNet mitigates computational complexity without compromising precision, ensuring optimal performance with reduced computational overhead. Leveraging the BRA algorithm, which features dynamic and sparse attention mechanisms, further enhances feature extraction by prioritizing salient features and minimizing redundancy. This strategic focus boosts the model's ability to discern crucial details within the input data. Moreover, the incorporation of a structured feature fusion network, based on GFPN, reconstructs the Neck component of YOLOv8, facilitating effective feature fusion across multiple scales and bolstering the model's capacity to extract comprehensive information from diverse contexts. Lastly, the introduction of an added detection head refines detection performance by incorporating additional anchor box scales and optimizing regression losses. This enhancement improves the model's ability to accurately localize and classify objects of interest accurately. Together, these components

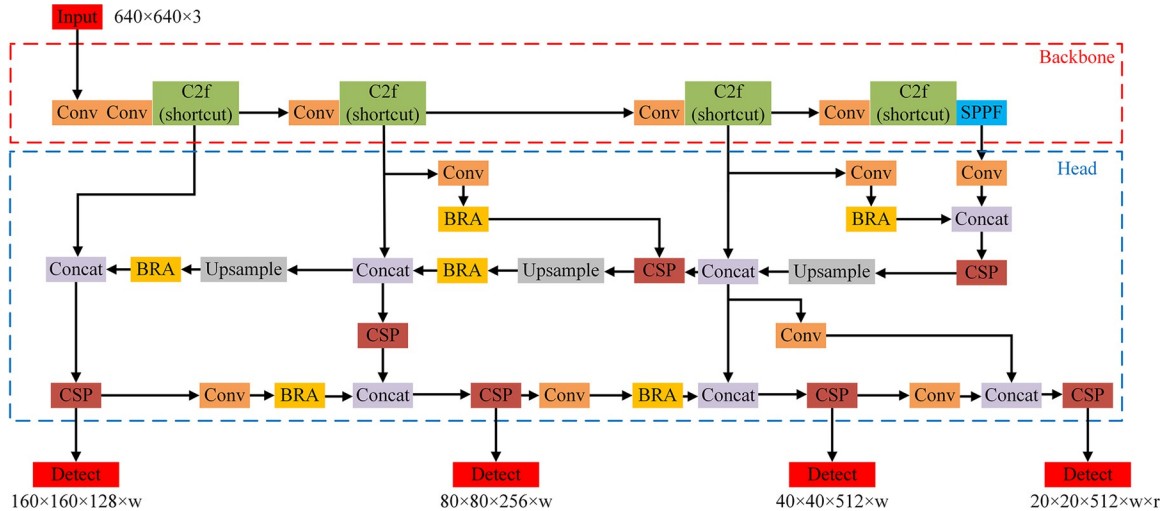

**Fig 2. Overall architecture diagram of RSG-YOLOV8 model.**

constitute the comprehensive architecture of the RSG-YOLOV8 model, poised to advance the state-of-the-art in object detection tasks.

*1.2.1 CSPDenseNet.* The Cross Stage Partial (CSP) Network, a modular design employed to enhance the performance of neural networks, is exemplified in the architecture of CSPDense-Net [7], as illustrated in Fig 3. By dividing the network into two distinct segments and incorporating partial connections between them, CSPDenseNet significantly enhances feature transfer and reuse. This configuration allows for a seamless exchange of information between stages, thus improving the feature representation capacity and overall efficacy of the model. Merging the dense connection structure of DenseNet with the innovative CSP module, CSPDenseNet represents a powerful synergy of complementary technologies.

Initially, a DenseNet backbone is established to leverage its dense connectivity. The CSP module is then strategically integrated into the mid-layers of this backbone, effectively splitting the network into two parts while maintaining partial connections between them. This design paradigm enables efficient information transmission, capitalizing on DenseNet's strengths while simultaneously improving feature reuse and information transmission through the CSP module. Particularly in the context of detecting germinated rice seeds, this combination of methodologies within CSPDenseNet is poised to significantly enhance performance metrics.

*1.2.2 BRA-based attention feature fusion.* The primary aim of the multi-scale feature fusion network, situated in the Neck section, is to amalgamate feature maps extracted from various layers of the network, thereby augmenting the efficacy of multi-scale object detection. However, the feature fusion layer in YOLOv8 encounters challenges related to redundant information from distinct feature mappings. To mitigate this limitation, this paper explores the incorporation of an attention mechanism within the feature fusion process of the YOLOv8 model. Attention mechanisms, initially proposed to evaluate the relevance of specific features over others, have shown substantial promise in enhancing object detection performance within computer vision.

Five attention mechanisms—Squeeze-and-Excitation (SE) [17], Convolutional Block Attention Module (CBAM) [18], Efficient Channel Attention (ECA) [19], Coordinate Attention (CA) [20], Receptive Field Attention (RFA) [21], and BRA—are identified as significantly

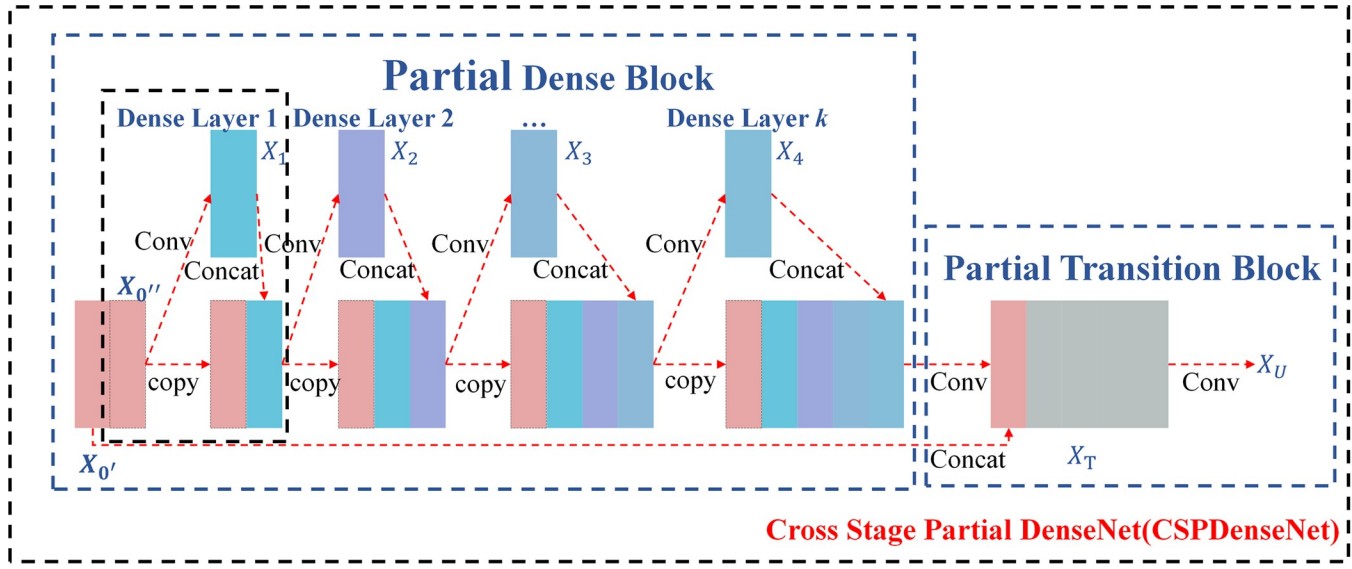

**Fig 3. Overall architecture diagram of CSPDenseNet.**

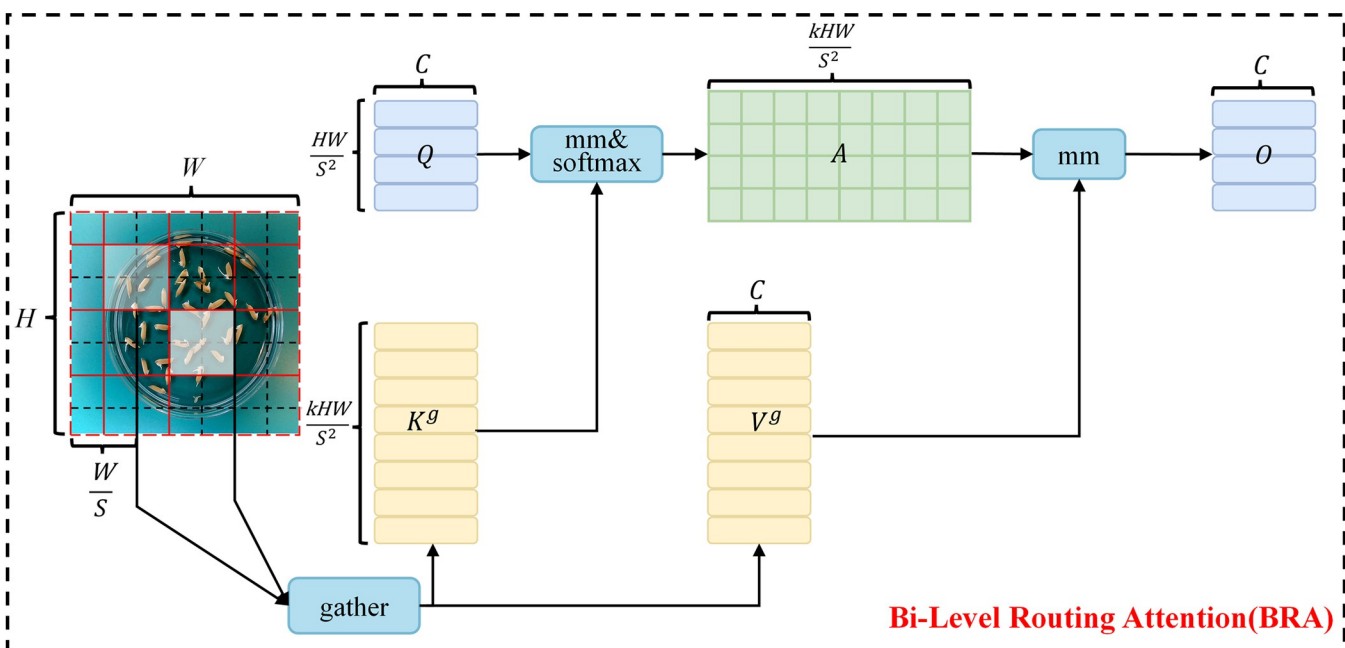

**Fig 4. Overall architecture diagram of BRA module.**

potential in improving object detection performance. Their distinctions arise from SE and ECA's concentration on channel attention, RFA and BRA's focus on spatial attention, while CBAM and CA enhance both channel and spatial attentions simultaneously. SE recalibrates feature responses at the channel level by explicitly modeling the interdependencies among convolutional feature channels. ECA captures local dependencies among channels with minimal computational demands, eliminating the reliance on global statistics. RFA introduces an effective attention mechanism for facilitating parameter sharing among convolutional kernels, whereas BRA embodies a dynamic, query-aware sparse attention mechanism, enabling a content-aware selection of the most relevant key/value tokens for each query.

Fig 4 illustrates the comprehensive architecture of the BRA module within RSG-YOLOV8. The feature fusion structure is enhanced through the integration of the BRA attention module, facilitating efficient multi-level feature fusion while mitigating redundant information across feature mappings. The introduction of dynamic sparse attention facilitates the integration of feature maps of varying scales by leveraging weight distributions of individual channels and spatial positions, thereby diminishing redundant feature information and enhancing the model's detection accuracy.

To handle a 2D input feature map $X \in R^{H \times W \times C}$, the initial step involves partitioning it into $S \times S$ non-overlapping regions, where each region contains $\frac{HW}{S^2}$ feature vectors. This partitioning is achieved by reshaping $X$ as $X^r \in R^{S^2 \times \frac{HW}{S^2} \times C}$. Subsequently, the query, key, and value tensors, $Q, K, V \in R^{S^2 \times \frac{HW}{S^2} \times C}$ are obtained, through linear projections:

$$Q = X^r W^q, K = X^r W^k, V = X^r W^v \qquad (1)$$

where $W^q, W^k, W^v \in R^{C \times C}$ represent the projection weights for the query, key, and value, respectively.

The attending relationship, determining which regions should be attended to for each given region, is established by constructing a directed graph. Initially, region-level queries and keys,

$Q^r$, $K^r \in R^{S^2 \times C}$ respectively, are derived by averaging $Q$ and $K$ per region. Subsequently, the adjacency matrix $A^r \in R^{S^2 \times S^2}$ of the region-to-region affinity graph is obtained through matrix multiplication between $Q^r$ and the transpose of $K^r$:

$$A^r = Q^r(K^r)^T \qquad (2)$$

Entries in the adjacency matrix $A^r$ quantify the semantic relationship between two regions. The subsequent crucial step involves pruning the affinity graph by retaining only the top-k connections for each region. This is achieved by deriving a routing index matrix $I_r \in N^{S^2 \times k}$ using the row-wise top-k operator:

$$I^r = topkIndex(A^r) \qquad (3)$$

Thus, the $i^{th}$ row of $I^r$ contains $k$ indices representing the most relevant regions for the $i^{th}$ region.

In the feature fusion process, RSG-YOLOV8 strategically places the BRA module subsequent to the Convolutional or Upsampling module, enabling the model to concentrate solely on specific regions post feature extraction. The primary objective of the BRA module is to eliminate irrelevant key-value pairs input at a broader regional level, retaining only pertinent domains. Initially, the BRA module takes the feature map as input, partitioning it into distinct regions and deriving queries, keys, and values through linear transformations.

Subsequently, the region-level relationship between queries and keys is fed into an adjacency matrix to construct a directed graph, precisely identifying the association of specific key-value pairs. This delineation determines the involvement of designated areas. Ultimately, utilizing the region-to-region routing index matrix facilitates multi-head self-attention between individual tokens. The double-layer path optimization of multi-head self-attention directs more attention towards the newly germinated part of the feature map of rice seeds, thereby augmenting the model's capability to detect germinated rice seeds.

The approach proposed in this paper solely employs the attention module BRA from BiFormer, distinguishing it from existing methodologies that incorporate BiFormer into YOLOv8 [8, 22–24].

*1.2.3 GFPN*. FPN successfully addresses the challenge of integrating hierarchical features into convolutional neural networks, thereby enhancing the performance of object detection models across various scales. PANet further refines feature propagation and information sharing within the feature pyramid. Introducing a bottom-up pathway alongside FPN's top-down approach, the Bi-directional Feature Pyramid Network (BiFPN) [25] effectively exploits multi-scale features. Conversely, GFPN implements dense connections and the Queen-Fusion with Concat technique to minimize information loss.

As shown in Fig 5, the Queen-fusion connection in $P_5$ consists of the previous layer $P_4$ downsampling, the previous layer $P_6$ upsampling, the previous layer $P_5$ and the current layer $P_4$. In this work, bilinear interpolation and maximum pooling are applied as the upsampling and downsampling functions, respectively. Therefore, in the case of extreme large-scale changes, the model is required to have sufficient exchange of information between the upper and lower layers. Based on the mechanism of layer-hopping and cross-scale connectivity, GFPN can be as long as the 'giraffe's neck'. With such a 'heavy neck' and a lightweight backbone, RSG-YOLOV8 can strike a balance between higher accuracy and better efficiency.

FPN and PANet play pivotal roles in facilitating multi-scale feature fusion within the Neck architecture of both YOLOv5 and YOLOv8. YOLOv8's Neck incorporates the C2f module instead of C3 during the upsampling process, distinguishing it from YOLOv5. FPN extracts

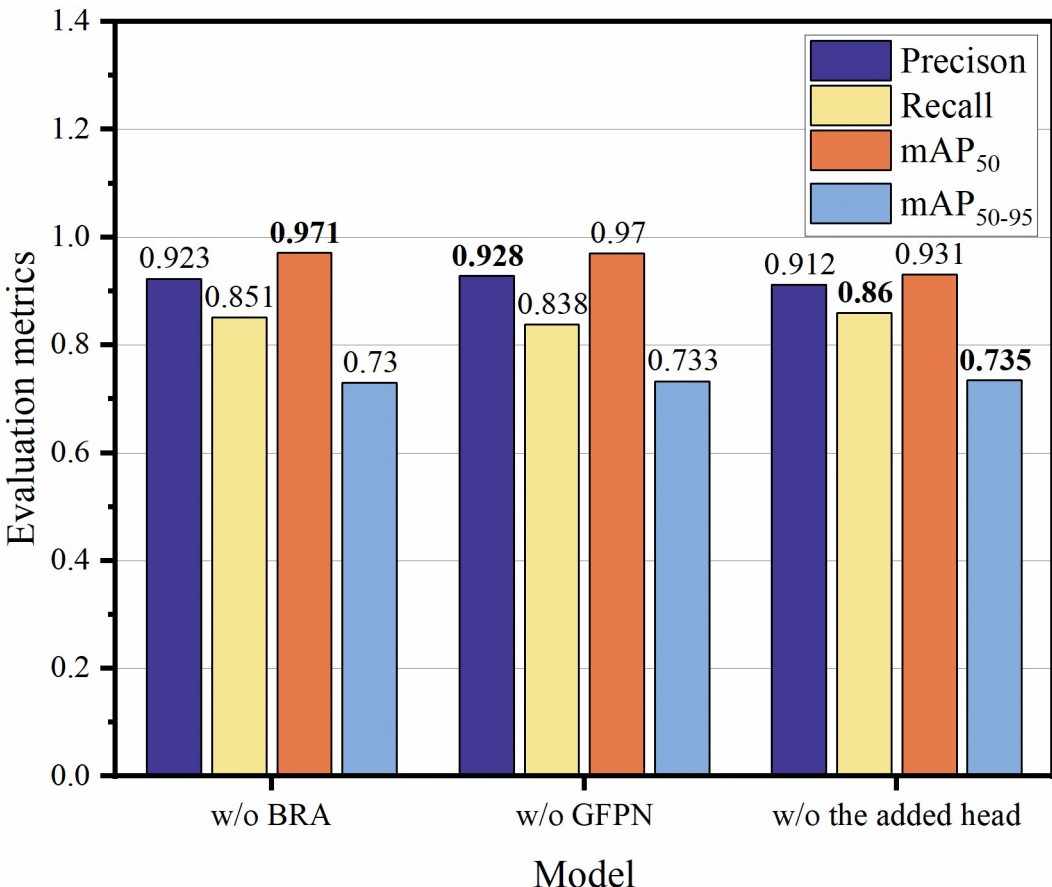

**Fig 5. Illustration of Queen-fusion in GFPN.** C represent concatenation fusion style, and 'Pk denotes node in next layer.

CNN feature maps and fuses them in a top-down manner with upsampling and coarse-grained maps, while PANet integrates bottom-up information to preserve spatial details. BiFPN, AFPN [26], and GFPN efficiently integrate features at different levels, thereby enhancing effectiveness by incorporating additional fusion levels. RSG-YOLOV8 enhances the FPN-PANet structure in YOLOv8 to improve multi-level feature fusion through multi-path network fusion [27].

*1.2.4 Added detection head.* The original YOLOv8 features three detection heads with dimensions of 20×20, 40×40, and 80×80 respectively. However, these heads fail to adequately meet the detection requirements in rice seed detection scenarios, resulting in suboptimal accuracy for objects exceeding the original scale. RSG-YOLOV8 addresses this limitation by introducing an additional detection head sized at 160×160 within the Head component. Moreover, it incorporates a novel feature fusion network structure within the Neck component to enhance detection capabilities across various object scales. Adjacent to the existing 80x80 detection scale in YOLOv8, the new detection head is added, integrating shallow information from the initial C2f (shortcut) module in the input image and combining it with an additional feature fusion network. This augmentation by RSG-YOLOV8 significantly improves the model's ability to detect larger objects. RSG-YOLOV8's architecture builds upon YOLOv8, integrating new modules such as BRA and CSP, alongside existing modules like Conv, C2f (shortcut), SPPF, Concat, Upsample, and Detect.

## 2. Dataset and parameter configuration

**2.1 Dataset.** The dataset utilized in this study is the open-source RiceSeedGermination dataset, sourced from the Kaggle official website (https://www.kaggle.com). Comprising rice seed images from nine distinct populations, this dataset showcases wide phenotypic diversity across different strains within each population, including variations in length, shape, and color. In total, the dataset encompasses 600 rice seed images, segregated into training and testing sets at an 8:2 ratio. Within these images, a diverse array of seeds is depicted, characterized by varying sizes, shapes, and colors, alongside incidental impurities such as branch stalks, broken leaves, and rice awns. Notably, these seeds are randomly distributed throughout the images.

**2.2 Parameter configuration.** RSG-YOLOV8 undergoes training and testing utilizing the Intel® Core® i5-10200H (2.40GHz) CPU and NVIDIA® GeForce GTX® 1650 12GB GPU. The software environment consists of the Windows version of PyCharm 2023.1. The proposed method is an implementation based on YOLOv8 architecture. Hyperparameters employed in training RSG-YOLOV8 and other comparative methods mirror those of YOLOv8. Training parameters are configured with a batch size of 1 and 300 epochs. The optimization process utilizes the AdamW optimizer with initial and final learning rate set to 0.0001, a momentum of 0.937, and a network input size of 640×640 [28].

## 3. Experiment and result analysis

**3.1 Performance indicators.** To objectively evaluate the effectiveness of the proposed method, the paper assesses the performance of RSG-YOLOV8 using four evaluation metrics: Precision, Recall, $mAP_{50}$, F1-score, and $mAP_{50-95}$. Precision and Recall are calculated using the following formulas:

$$Precision = \frac{TP}{TP + FP} \tag{4}$$

$$Recall = \frac{TP}{TP + FN} \tag{5}$$

$$F1 - score = 2 \times \frac{Precision \times Recall}{Precision + Recall} \tag{6}$$

where *TP* represents the number of samples correctly classified as positive, *FP* indicates the number of samples incorrectly classified as positive, and *FN* represents the number of samples incorrectly classified as negative.

$$AP = \int_0^1 P(R) \tag{7}$$

$$mAP = \frac{1}{C} \sum_{i=1}^{C} AP_i \tag{8}$$

$$mAP_{50} = \frac{1}{C} \sum_{i=1}^{C} AP_{50_i} \tag{9}$$

$$mAP_{50-95} = \frac{1}{C}\sum_{i=1}^{C} AP_{50-95_i} \tag{10}$$

where *AP* denotes the area under the precision–recall curve for a specific category at various confidence thresholds. *mAP* represents the mean average precision, calculated by averaging the *AP* values across all categories. $mAP_{50}$ refers to the *mAP* computed with an IOU threshold of 0.5, while $mAP_{50-95}$ refers to the *mAP* calculated with an IOU threshold ranging from 0.5 to 0.95, providing a more stringent evaluation metric.

**3.2 Ablation experiments.**   To ascertain the efficacy of the proposed strategy in augmenting the performance of deep neural networks, this paper conducts an exhaustive series of ablation experiments. These experiments were meticulously crafted to gauge the individual impact of each component within the model on the overall performance metrics. Through this systematic evaluation, this paper aimed to elucidate the theoretical underpinnings and practical advantages of the proposed solution.

*3.2.1 Ablation experiments of the overall structure*. Through the removal of each component, the study assessed four incomplete RSG-YOLOV8 models. As illustrated in Fig 6, it is evident that BRA, GFPN, the added detection head, and Generalized Intersection over Union (GIoU) [29] all contribute to enhancing the accuracy of RSG-YOLOV8. 'w/o GFPN' denotes the utilization of the original Neck structure, FPN-PANet, from YOLOv8. Notably, the inclusion of the added detection head yields the most substantial improvement in overall accuracy, particularly for $mAP_{50}$, followed by GFPN and BRA.

*3.2.2 Comparison of different multi-scale feature fusion structures*. This paper presents a comparative analysis of the proposed RSG-YOLOV8 with BRSG-YOLOV8 and ARSG-YOLOV8. In BRSG-YOLOV8 and ARSG-YOLOV8, the GFPN in the Neck component of RSG-YOLOV8 is substituted with BiFPN and AFPN for feature fusion. As depicted in Fig 7, RSG-YOLOV8 utilizing the GFPN structure exhibits significantly higher $mAP_{50}$ and Recall metrics compared to models employing BiFPN and AFPN structures.

*3.2.3 Comparison of different attention mechanisms*. This study employed the RSG-YOLOV8 model to investigate various attention mechanisms. The initials of the model names outlined in Table 1 denote the attention mechanisms utilized, specifically S, E, C, A, R, and B, representing SE, ECA, CBAM, CA, RFA, and BRA, respectively. Among the alternative attention mechanisms considered, BRA yielded the most substantial performance enhancement. Additionally, CBAM (i.e. CRSG-YOLOV8) demonstrated an $mAP_{50}$ second only to BRA (i.e. RSG-YOLOV8), with its accuracy values surpassing those of BRA. Despite ECA (i.e. ERSG-YOLOV8) exhibiting a higher $mAP_{50-95}$ compared to BRA, its $mAP_{50}$ notably lagged behind that of BRA.

*3.2.4 Comparison of different regression losses*. This study conducted ablation experiments to assess the impact of various regression losses in object detection, including GIoU, which measures the distance between two rectangles aligned along both axes, Distance-IoU (DIoU), Effective-IoU (EIoU), Scylla-IoU (SIoU) [30], and Wise-IoU (WIoU)v3 [23, 31]. These loss functions are denoted by the added letter of the model names in Table 2: G, D, E, S, W. In comparison to other regression losses, the original regression loss, Complete Intersection over Union (CIoU) in YOLOv8, demonstrates superior robustness for bounding boxes.

The $mAP_{50}$ of DIoU (RSGD-YOLO) closely aligns with that of CIoU (RSG-YOLOV8), suggesting DIoU as a viable competitor to CIoU. Conversely, for $mAP_{50-95}$, GIoU (RSGG-YOLO) outperforms CIoU. The choice of regression loss depends on specific scenario criteria. In this

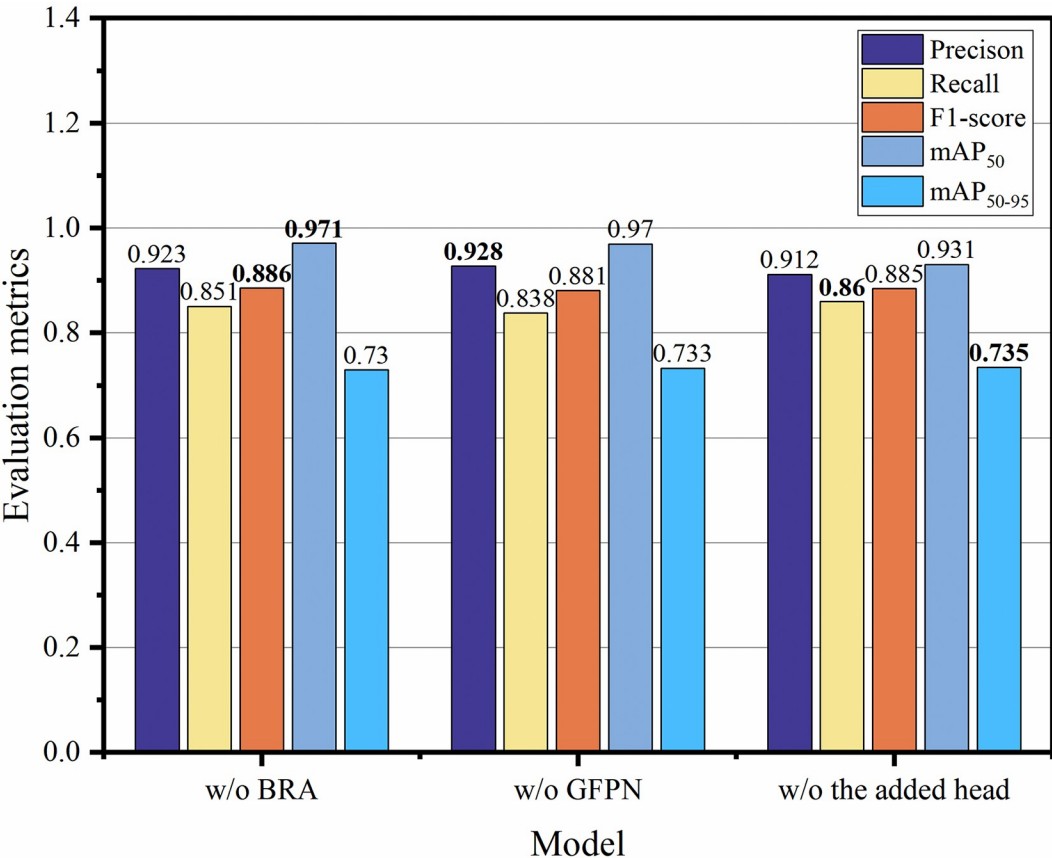

**Fig 6. Ablation experiments of each method in the proposed RSG-YOLOV8.** 'w/o' represents without.

study, $mAP_{50}$ serves as the primary indicator for detecting the germination rate of rice seeds. Consequently, CIoU is selected as the regression loss in the proposed RSG-YOLOV8.

**3.3 Comparison of the results of different models.** In order to make a fair comparison, this study chose the competitive model and evaluated them using the same evaluation indicators. As shown in Table 3, compared to YOLOv8, RSG-YOLOV8 has improved by 0.5%, 7.1%, 4%, and 5.8% in Precision, Recall, $mAP_{50}$, and $mAP_{50-95}$, respectively. RSG-YOLOV8 not only surpasses the baseline YOLOv8 model, but also outperforms YOLOv5, YOLO-r, and YOLOv7 model.

Fig 8 illustrates the dynamic trends of various evaluation metrics throughout the training process of the YOLOv5, YOLOv7, YOLOv8, YOLO-r, and RSG-YOLOV8 models.

In Fig 8(A), the fluctuation of Precision among the models with epochs is depicted. Notably, RSG-YOLOV8 initially exhibits lower Precision, gradually ascending to its peak at the 300th epoch. YOLOv7 and YOLOv8 demonstrate parallel increases in Precision, with YOLOv8 surpassing YOLOv7 in later epochs. YOLOv5, represented by the black line, initially presents lower Precision, with marginal improvement over time and YOLO-r model has the fastest convergence speed.

Fig 8(B) showcases the evolution of Recall over epochs, mirroring the trends observed in Precision. RSG-YOLOV8 initiates with a subdued Recall, progressively augmenting thereafter. Similarly, YOLOv7, YOLO-r, and YOLOv8 exhibit consistent growth in Recall. Conversely, YOLOv5 demonstrates a more stabilized Recall progression.

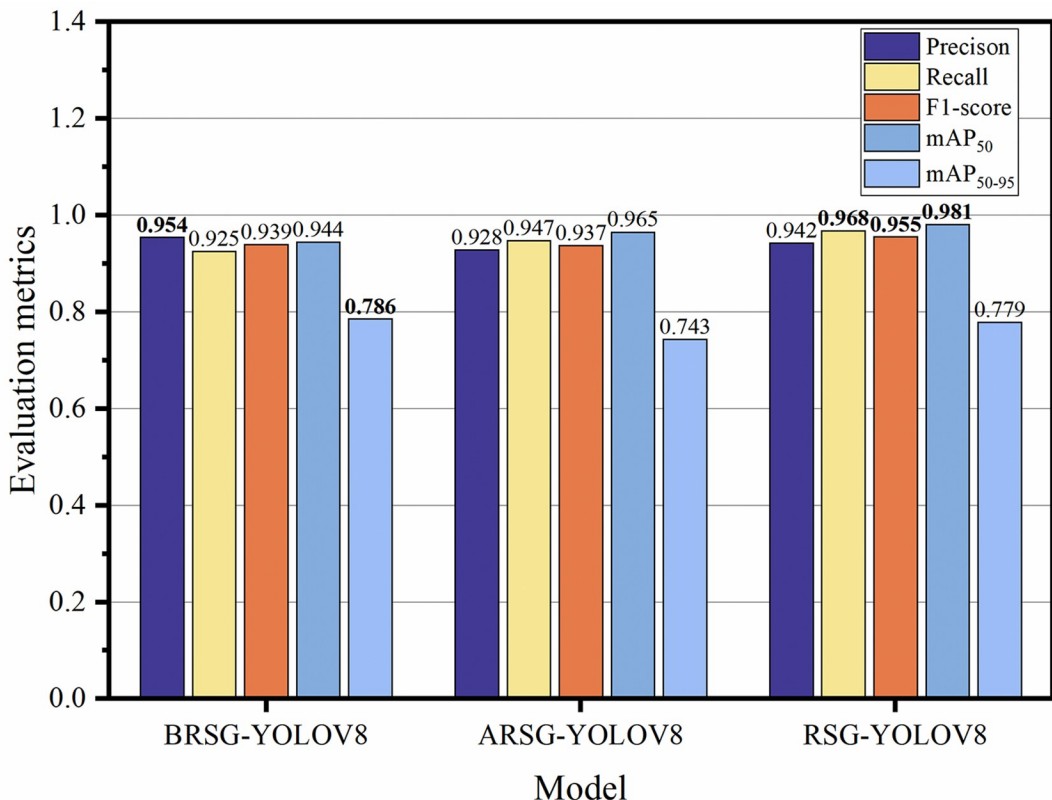

**Fig 7. Ablation experiments of multi-scale feature fusion structure.** Replace the GFPN structure in the Neck part of RSG-YOLOV8 with BiFPN and AFPN. The best results are displayed in bold.

Fig 8(C) delineates the variations in $mAP_{50}$ over epochs. Initially, RSG-YOLOV8 displays a notably high $mAP_{50}$, which continues to ascend steadily throughout the training duration, culminating in a significantly elevated level. While YOLOv8's initial $mAP_{50}$ slightly lags behind YOLOv7, it experiences rapid enhancement, eventually either matching or surpassing YOLOv7's $mAP_{50}$ in later epochs. YOLOv5 commences with a slightly higher initial $mAP_{50}$, yet its growth rate decelerates over subsequent epochs.

Fig 8(D) illustrates the dynamics of $mAP_{50-95}$ across epochs. Initially, with fewer epochs, minimal discrepancies in $mAP_{50-95}$ are discernible among the models; however, as epochs progress, distinctions become more pronounced. The blue line representing YOLOv8 depicts a relatively erratic $mAP_{50-95}$ trend throughout iterations, gradually improving with increasing epochs. Notably, RSG-YOLOV8 exhibits exceptional $mAP_{50-95}$ performance with a larger

**Table 1. Ablation experiments of attention mechanisms.** Replace BRA in RSG-YOLOV8 with SE, ECA, CBAM, CA, RFA respectively. The best results are highlighted in bold.

| Model | Precision | Recall | F1-score | $mAP_{50}$ | $mAP_{50-95}$ |
|---|---|---|---|---|---|
| SRSG-YOLOV8 | 0.918 | 0.864 | 0.890 | 0.954 | 0.650 |
| ERSG-YOLOV8 | 0.942 | 0.875 | 0.907 | 0.944 | 0.782 |
| CRSG-YOLOV8 | **0.959** | 0.856 | 0.905 | 0.976 | 0.672 |
| ARSG-YOLOV8 | 0.936 | 0.873 | 0.903 | 0.958 | 0.752 |
| RRSG-YOLOV8 | 0.905 | 0.928 | 0.916 | 0.945 | 0.772 |
| **RSG-YOLOV8** | 0.942 | **0.968** | **0.955** | **0.981** | **0.779** |

**Table 2. Ablation experiments of regression loss.** Replacing the CIoU in RSG-YOLOV8 with GIoU, DIoU, EIoU, SIoU, and WIoU, respectively. The best results are highlighted in bold.

| Model | Precision | Recall | F1-score | mAP$_{50}$ | mAP$_{50\text{-}95}$ |
|---|---|---|---|---|---|
| RSGG-YOLO | **0.955** | 0.902 | 0.928 | 0.965 | 0.783 |
| RSGD-YOLO | 0.907 | 0.914 | 0.910 | 0.976 | **0.781** |
| RSGE-YOLO | 0.935 | 0.919 | 0.927 | 0.958 | 0.761 |
| RSGS-YOLO | 0.904 | 0.856 | 0.879 | 0.946 | 0.775 |
| RSGW-YOLO | 0.918 | 0.943 | 0.930 | 0.960 | 0.755 |
| **RSG-YOLOV8** | 0.942 | **0.968** | **0.955** | **0.981** | 0.779 |

number of epochs, surpassing its counterparts. While the mAP$_{50\text{-}95}$ curves of YOLOv5 and YOLOv7 remain closely aligned, as epochs accumulate, YOLOv7 marginally outperforms YOLOv5 in terms of mAP$_{50\text{-}95}$.

Fig 8(E) shows the dynamics of F1-score over epochs. YOLOv5's F1-score rises smoothly with increasing epochs, demonstrating a stable training process; YOLOv7 is similar to YOLOv5, but has a faster performance improvement rate, reaching higher F1-score at certain epoch points; RSG-YOLOV8 does not have a high F1-score at the beginning, but with increasing epochs, the F1-score increases significantly, eventually surpassing other YOLO series models; YOLOv8 maintains high F1-score throughout the training process, showing good stability and generalisation ability; YOLO-r has a more fluctuating trend in F1-score, but also improves with epochs increase in general, but not as significantly as the other models.

Fig 9 illustrates the performance comparison among YOLOv5, YOLOv7, YOLOv8, and RSG-YOLOV8 models in germinated rice seed detection, with red prediction boxes denoting germinated rice seed labeled as 'yes.' The images provide a clear depiction of each model's performance on the same scene and object. Fig 9(A) and 9(G) depict original images randomly selected from the dataset. Fig 9(C) and 9(I) showcase the results recognized by YOLOv7, which already demonstrate the identification of some germinated rice seeds with relatively clear bounding boxes. Fig 9(B) and 9(H) present the outcomes of YOLOv5 recognition, showcasing further improvement compared to YOLOv7 in both object recognition and bounding box accuracy. Fig 9(D) and 9(J) display the recognition results of YOLOv8, which significantly elevate recognition effectiveness. Nearly all germinated rice seeds are accurately identified, with bounding box accuracy approaching perfection, aligning closely with the actual seed boundaries. Fig 9(E) and 9(K) showcase the results recognized by YOLO-r, which although identifies many germinated seeds, there are still false detections and missed detections. Lastly, Fig 9(F) and 9(L) demonstrate the results of RSG-YOLOV8 recognition, representing the latest technological advancement among these versions, and showcasing exceptional performance. In terms of recognition accuracy, it approaches near perfection, successfully identifying all objects while achieving the ultimate precision in bounding box delineation.

**Table 3. Performance comparison of YOLOv5, YOLOv7, YOLOv8, YOLO-r, and RSG-YOLOV8 models.** The best results are highlighted in bold.

| Model | Precision | Recall | F1-score | mAP$_{50}$ | mAP$_{50\text{-}95}$ |
|---|---|---|---|---|---|
| YOLOv5 | 0.907 | 0.929 | 0.918 | 0.961 | 0.677 |
| YOLOv7 | 0.797 | 0.835 | 0.816 | 0.857 | 0.693 |
| YOLOv8 | 0.937 | 0.897 | 0.917 | 0.941 | 0.721 |
| YOLO-r | 0.925 | 0.967 | 0.946 | 0.925 | 0.620 |
| **RSG-YOLOV8** | **0.942** | **0.968** | **0.955** | **0.981** | **0.779** |

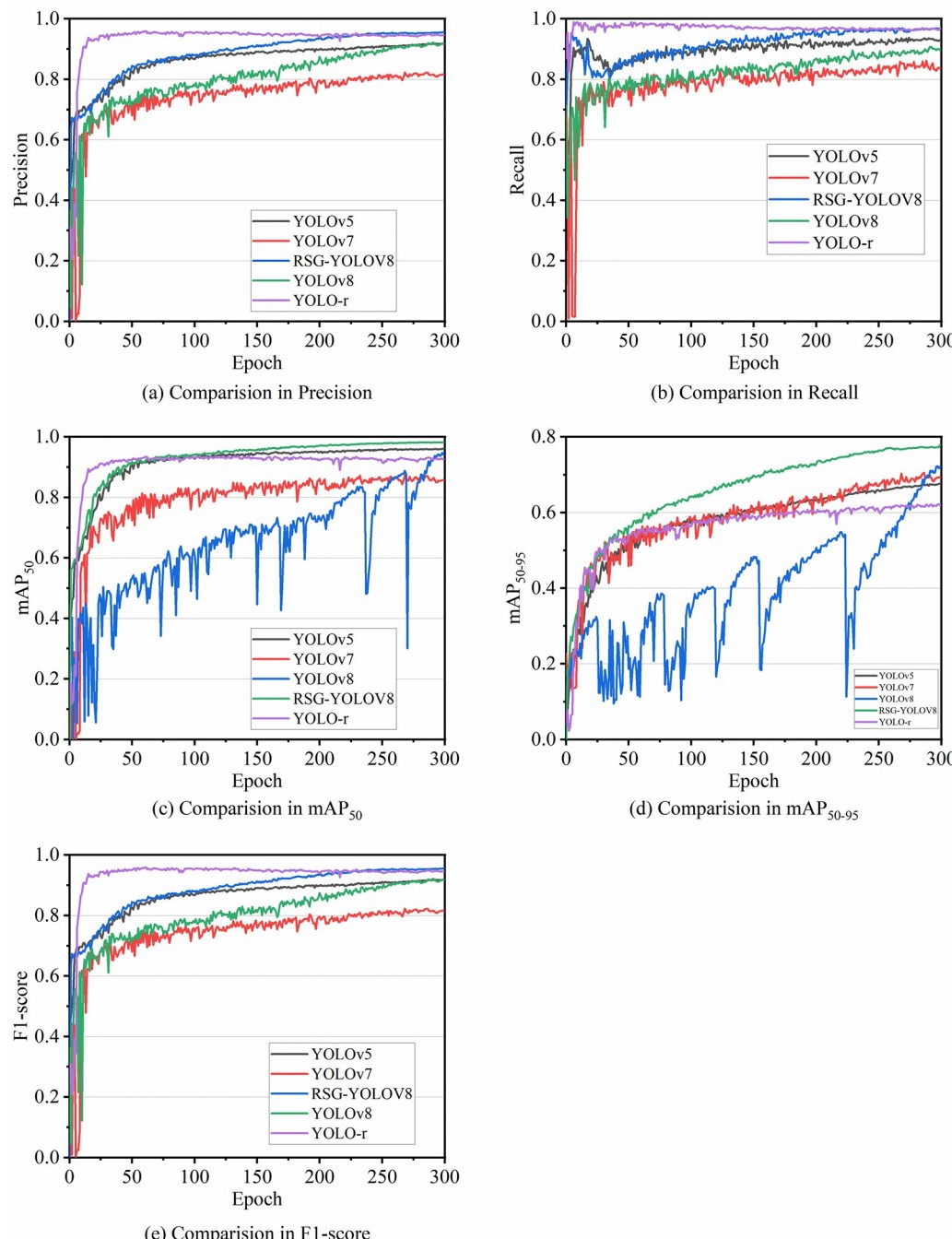

**Fig 8. Change curves of evaluation metrics of YOLOv5, YOLOv7, YOLOv8, YOLO-r, and RSG-YOLOV8 models.**

## Conclusion

The RSG-YOLOV8 model, a novel variant of YOLOv8 tailored specifically for precise detection of rice seed germination in images, is introduced in this paper. Through meticulous optimization of the GFPN feature fusion structure and incorporation of the BRA attention mechanism, along with the addition of a detection head, the target detection capabilities of YOLOv8 are markedly enhanced by RSG-YOLOV8. These enhancements facilitate weighted

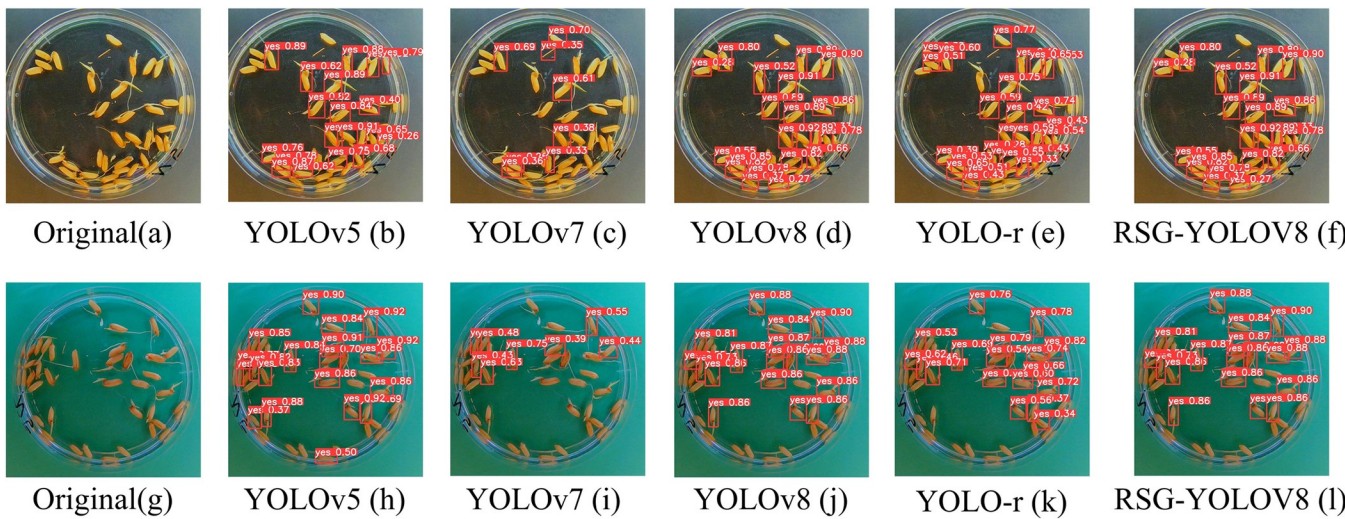

**Fig 9. Comparison of detection results between YOLOv5, YOLOv7, YOLOv8, and RSG-YOLOV8 models.**

feature fusion across multiple levels and diverse scales, culminating in the generation of high-quality anchor boxes with a dynamic focusing mechanism. The effectiveness of the proposed algorithm is rigorously validated using the RiceSeedGermination dataset. Experimental results demonstrate the superior detection accuracy of the RSG-YOLOV8 algorithm, with an $mAP_{50}$ of 0.981, representing a noteworthy 4% improvement over the original YOLOv8 model. Furthermore, RSG-YOLOV8 surpasses alternative comparative experiments, achieving a Recall of 0.968, which is notably 7.1% higher than the baseline YOLOv8 model without any enhancements.

## Author Contributions

**Conceptualization:** Yujun Zhang.

**Data curation:** Huikang Li, Lu Liu.

**Formal analysis:** Yujun Zhang.

**Funding acquisition:** Longbao Liu.

**Methodology:** Huikang Li.

**Project administration:** Qixing Tang, Yuan Rao, Yanwei Gao.

**Resources:** Yujun Zhang.

**Software:** Lu Liu.

**Supervision:** Qi Li, Juan Liao.

**Validation:** Huikang Li, Qi Li, Juan Liao.

**Writing – original draft:** Huikang Li.

**Writing – review & editing:** Longbao Liu.

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
