## [Decision Letter · Decision Letter 0]

28 Aug 2024

PONE-D-24-24638RSG-YOLO: Detection of rice seed germination rate based on enhanced YOLOv8 and multi-scale attention feature fusionPLOS ONE

Dear Dr. Tang,

Thank you for submitting your manuscript to PLOS ONE. After careful consideration, we feel that it has merit but does not fully meet PLOS ONE’s publication criteria as it currently stands. Therefore, we invite you to submit a revised version of the manuscript that addresses the points raised during the review process.

We look forward to receiving your revised manuscript.

Kind regards,

Luca Bertolaccini, M.D., Ph.D.

Academic Editor

PLOS ONE

Journal Requirements:

6. PLOS requires an ORCID iD for the corresponding author in Editorial Manager on papers submitted after December 6th, 2016. Please ensure that you have an ORCID iD and that it is validated in Editorial Manager. To do this, go to ‘Update my Information’ (in the upper left-hand corner of the main menu), and click on the Fetch/Validate link next to the ORCID field. This will take you to the ORCID site and allow you to create a new iD or authenticate a pre-existing iD in Editorial Manager.

Additional Editor Comments:

The reviewers have emphasised issues that require a careful and thorough manuscript revision.

No commitment to publication can be made at this point.

Reviewers' comments:

Reviewer's Responses to Questions

**Comments to the Author**

1. Is the manuscript technically sound, and do the data support the conclusions?

Reviewer #1: Yes

Reviewer #2: Yes

2. Has the statistical analysis been performed appropriately and rigorously? 

Reviewer #1: No

Reviewer #2: Yes

3. Have the authors made all data underlying the findings in their manuscript fully available?

Reviewer #1: No

Reviewer #2: Yes

4. Is the manuscript presented in an intelligible fashion and written in standard English?

Reviewer #1: Yes

Reviewer #2: Yes

5. Review Comments to the Author

Reviewer #1: The paper presents a novel RSG-YOLO model with promising enhancements. Please clarify the below points to strengthen the manuscript's contribution

1.Can you provide more details on the Rice Seed Germination dataset used for validation? How was the dataset curated to ensure its representativeness and reliability for training and evaluating the RSG-YOLO model?

2.The BRA module is described as a dynamic and sparse attention mechanism. Can you provide quantitative details on how BRA compares to other attention mechanisms (e.g., SE, CBAM, ECA) in terms of performance improvements and computational overhead? How does BRA’s dynamic and sparse nature contribute to the reduction of redundancy and enhancement of critical feature highlighting?

3.How does the GFPN-based structured feature fusion network improve feature integration compared to the original Neck component of YOLOv8? Please detail the advantages of GFPN’s approach in handling multi-scale features and its impact on overall model performance.

4.The introduction of an additional detection head is mentioned. What specific improvements in detection performance (e.g., precision, recall) were observed with this modification? How do variable anchor box scales and optimized regression losses contribute to these performance improvements?

5.What specific steps were taken to ensure that RSG-YOLO performs robustly under varying conditions such as different lighting scenarios or camera angles? How does the model's performance vary across these conditions?

6.Can you elaborate on how the integration of CSP DenseNet and the BRA attention mechanism interacts with the YOLOv8 architecture to enhance detection performance? How do these architectural changes affect the trade-offs between computational complexity and accuracy?

Reviewer #2: The manuscript can be published with the following changes:

1. The version of yolo modified can be mentioned in proposed method name also.

2. Block diagram of the proposed system needs to be added.

3. why F1 score is not used for evaluation justify.

4. The research gap leading to methodology needs to be mentioned at the end of literature review.

5. The following references can be added

i) Venkatesan, R., & Balaji, G. N. (2024). Balancing composite motion optimization using R-ERNN with plant disease. Applied Soft Computing, 154, 111288

ii) Appe, S. N., Arulselvi, G., & Balaji, G. N. (2023). Detection and Classification of Dense Tomato Fruits by Integrating Coordinate Attention Mechanism With YOLO Model. In Handbook of Research on Deep Learning Techniques for Cloud-Based Industrial IoT (pp. 278-289). IGI Global.

6. PLOS authors have the option to publish the peer review history of their article (what does this mean?). If published, this will include your full peer review and any attached files.

Reviewer #1: No

Reviewer #2: No

---

## [Author Response · Author response to Decision Letter 0]

25 Sep 2024

Dear Editor and Reviewers, the Respond to Reviewers has been included in the Revision Report.

---

## [Decision Letter · Decision Letter 1]

18 Oct 2024

RSG-YOLOV8: Detection of rice seed germination rate based on enhanced YOLOv8 and multi-scale attention feature fusion

PONE-D-24-24638R1

Dear Dr. Tang,

We’re pleased to inform you that your manuscript has been judged scientifically suitable for publication and will be formally accepted for publication once it meets all outstanding technical requirements.

Kind regards,

Luca Bertolaccini, M.D., Ph.D.

Academic Editor

PLOS ONE

Additional Editor Comments (optional):

Reviewers' comments:

Reviewer's Responses to Questions

**Comments to the Author**

1. If the authors have adequately addressed your comments raised in a previous round of review and you feel that this manuscript is now acceptable for publication, you may indicate that here to bypass the “Comments to the Author” section, enter your conflict of interest statement in the “Confidential to Editor” section, and submit your "Accept" recommendation.

Reviewer #1: All comments have been addressed

Reviewer #2: All comments have been addressed

2. Is the manuscript technically sound, and do the data support the conclusions?

Reviewer #1: Yes

Reviewer #2: Yes

3. Has the statistical analysis been performed appropriately and rigorously? 

Reviewer #1: Yes

Reviewer #2: Yes

4. Have the authors made all data underlying the findings in their manuscript fully available?

Reviewer #1: No

Reviewer #2: Yes

5. Is the manuscript presented in an intelligible fashion and written in standard English?

Reviewer #1: Yes

Reviewer #2: Yes

6. Review Comments to the Author

Reviewer #1: The article is well written, treats an actual problem and all the suggestions are addressed by the authors.

Reviewer #2: The authors made all the required changes, and the manuscript can be considered for publication. some grammatical errors can be checked if required.

7. PLOS authors have the option to publish the peer review history of their article (what does this mean?). If published, this will include your full peer review and any attached files.

Reviewer #1: No

Reviewer #2: No

---

## [Editor Report · Acceptance letter]

1 Nov 2024

PONE-D-24-24638R1 

PLOS ONE

Dear Dr. Tang, 

I'm pleased to inform you that your manuscript has been deemed suitable for publication in PLOS ONE. Congratulations! Your manuscript is now being handed over to our production team.

Kind regards, 

on behalf of

Dr. Luca Bertolaccini 

Academic Editor

PLOS ONE